# Investigation of commonly used aortic aneurysm growth rate metrics: Comparing their suitability for clinical and research applications

**Kayley Abell-Hart**[1], **Janos Hajagos**[1], **Victor Garcia**[1], **James Kaan**[2], **Wei Zhu**[3], **Mary Saltz**[1], **Joel Saltz**[1]*, **Apostolos Tassiopoulos**[2]

**1** Department of Biomedical Informatics, Stony Brook University, Stony Brook, NY, United States of America, **2** Department of Vascular Surgery, Stony Brook University Hospital, Stony Brook, NY, United States of America, **3** Department of Applied Mathematics and Statistics, Stony Brook University, Stony Brook, NY, United States of America

* joel.saltz@stonybrookmedicine.edu

**Data Availability Statement:** The data used in this paper is clinical data and cannot be made public due to IRB rules, Stony Brook institutional policies and US HIPAA regulations. Requests for data

## Abstract

An aneurysm is a pathological widening of a blood vessel. Aneurysms of the aorta are often asymptomatic until they rupture, killing approximately 10,000 Americans per year. Fortunately, rupture can be prevented through early detection and surgical repair. However, surgical risk outweighs rupture risk for small aortic aneurysms, necessitating a policy of surveillance. Understanding the growth rate of aneurysms is essential for determining appropriate surveillance windows. Further, identifying risk factors for fast growth can help identify potential interventions. However, studies in the literature have applied many different methods for defining the growth rate of abdominal aortic aneurysms. It is unclear which of these methods is most accurate and clinically meaningful, and whether these heterogeneous methodologies may have contributed to the varied results reported in the literature. To help future researchers best plan their studies and to help clinicians interpret existing studies, we compared five different models of aneurysmal growth rate. We examined their noise tolerance, temporal bias, predictive accuracy, and statistical power to detect risk factors. We found that hierarchical mixed effects models were more noise tolerant than traditional, unpooled models. We also found that linear models were sensitive to temporal bias, assigning lower growth rates to aneurysms that were detected earlier in their course. Our exponential mixed model was noise-tolerant, resistant to temporal bias, and detected the greatest number of clinical risk factors. We conclude that exponential mixed models may be optimal for large studies. Because our results suggest that choice of method can materially impact a study's findings, we recommend that future studies clearly state the method used and demonstrate its appropriateness.

access should be made to David Cyrille, Stony Brook Chief Research Information Officer - David.Cyrille@stonybrookmedicine.edu.

**Funding:** KAH This work was supported by grant T32GM127253 from the Scholars in BioMedical Sciences Training Program and the Science Training and Research to Inform Decisions fellowship via the National Science Foundation Research Traineeship program. The funders had no role in study design, data collection and analysis, decision to publish, or preparation of the manuscript.

**Competing interests:** The authors have declared that no competing interests exist.

## Introduction

An aortic aneurysm (AA) is a pathological widening of the body's largest artery. This condition is common, with abdominal aortic aneurysms (AAAs) present in about 4–7% of men over age 50, and about 1% of such women [1]. AAAs are often asymptomatic until rupture, a catastrophic event that is fatal in at least 75% of cases [1]. In 2018, nearly 10,000 people died from AAs in the United States [2]. Because rupture can be prevented through surgical repair of the aneurysm, screening programs have been deemed cost-effective in high-risk groups [1, 3]. However, surgery carries a risk of complications and mortality. For this reason, patients are not indicated for repair until their risk of rupture exceeds the risks associated with surgery [4].

The main predictor of rupture risk is size; therefore, the threshold for surgical intervention is largely dictated by the maximum diameter of the AA [5]. However, most newly discovered abdominal aortic aneurysms are small, and it is unknown when, if ever, a patient's aneurysm will cross the threshold for intervention. Therefore, standard of care requires regular imaging appointments to monitor the size of the AAA [3, 6, 7]. Choosing optimal surveillance windows is complicated by the fact that the rate of AAA growth can vary markedly among individuals. If the AAA will grow faster than expected, standardized surveillance windows may permit unchecked growth and elevated risk of rupture and death. If the AAA will grow more slowly than expected or even stabilize, the standard surveillance windows may be unnecessarily burdensome. In order to optimize the surveillance windows, various attempts have been made to predict AAA growth rate. An additional goal of such predictions is the potential identification of intervenable risk factors. For example, the observation that diabetes predicts slower AAA growth has led to a trial using the diabetes drug metformin to slow the growth of AAAs [8].

In order to predict growth rate or determine the risk factors associated with growth rate, it is first necessary to select a meaningful mathematic definition of growth rate itself. In the case of AAAs, this process is non-trivial. Because AAAs are known to grow faster at larger sizes, non-linear models may be needed. However, real-world datasets may contain only a few measurements of the aortic diameter per patient, cover a short time period, and/or be subject to substantial noise due to the imprecision of ultrasound measurements. Fitting a curve to a small number of unreliable datapoints may lead to extreme estimates of growth rate that are not well-justified. Previous studies have taken a variety of approaches. Many studies have simply taken the linear slope between each patient's first and last measurement, i.e. (last diameter– first diameter)/(last measurement time–first measurement time), disregarding any measurements in-between [6, 9–14]. Some studies have applied linear regression, using a patient's full set of observations but still applying a linear model [15–19]. Other studies have addressed the problem of limited data by applying hierarchical mixed models, which fit parameters to each patient while simultaneously maintaining a distribution of all the patients' parameters. This parent distribution is constrained by meta-parameters, which are derived from all patients. The hierarchical structure fits individual patient parameters while drawing statistical inference from the group, especially in cases with less data to justify an extreme or outlier parameter. This method is sometimes called partial pooling, because it can be seen as an intermediate between the unpooled approach, where each patient is analyzed separately, and a fully pooled approach, where all datapoints are combined and fit to a single line or curve [20].

Existing studies have applied linear, quadratic, and exponential hierarchical models to the AAA growth problem [7, 21–24]. Further clouding the question of an appropriate growth metric, some studies did not appear to specify the exact method used [25–27]. Different studies have produced widely varying estimates of AAA growth rate, even for similar initial diameters, and have produced a variety of findings regarding risk factors; a meta-analysis found studies reporting average AAA growth rates as low as -0.33 mm/yr and as high as 3.95 mm/yr [5]. It is

unclear which method might be most appropriate for making predictions or comparisons between patients to identify risk factors. Thus, there is a need for comparisons of the different methods side-by-side on the same dataset. We therefore present this investigation of AAA growth metrics. We investigate the stability, temporal bias, accuracy in prediction, and inferred risk factors across the first point—last point method, unpooled linear regression, unpooled exponential regression, a linear hierarchical mixed model, and an exponential hierarchical mixed model.

## Methods

### Dataset curation and preprocessing

Our dataset comes from an aortic aneurysm surveillance program at Stony Brook University Hospital. Patients with ectatic or aneurysmal aortas have been entered into our records and prospectively monitored to ensure timely follow-up care. This study was approved by the Stony Brook Internal Review Board. Because study activities were limited to records review, the review board waived consent requirements for this study. The master dataset from the surveillance program contained 943 patients with ectasia or aneurysm of the abdominal aorta. Patients were considered aneurysmal if the diameter of the abdominal aorta was three centimeters or greater, and ectatic if the diameter was 2.5 to 2.9 centimeters. In an oval shaped aorta, the long and short axis diameters are measured in multiple points. The short axis measurement in the location with the largest diameters is considered the maximum aneurysm diameter; this measurement is referred to as "AAA size" in this study. 58% of the measurements were taken from computerized tomography (CT) scans, and 41% were taken from ultrasound imaging. About 1% of measurements were taken from magnetic resonance imaging (MRI) or positron emission tomography (PET) scans.

In order to prevent extreme slopes from close-together measurements, we averaged together datapoints from the same patient that were less than 150 days apart. Details of how the dataset was filtered for each experiment are shown in Fig 1. To assess risk factors associated with clinical variables, we queried patient hospital data from a clinical database extracted from the electronic health records and mapped to the Observational Medical Outcomes Partnership (OMOP) Common Data Model [28]. A set of diseases, lab tests, medications, and demographic variables were selected by a clinician, along with the defining codes. Comorbid diagnoses were determined by mapping to Clinical Classifications Software (CCS) groupings [29]. Laboratory values were identified by Logical Observation Identifiers Names and Codes (LOINC), OMOP, and Systemized Nomenclature of Medicine (SNOMED) codes [30, 31]. Medications were selected by Anatomical Therapeutic Chemical (ATC) codes [32]. History of procedures was assessed by the International Classification of Diseases 10 Procedure Coding System (ICD-10-PCS) codes and by regular expressions (regex) [33]. Tobacco history was considered positive if a relevant ICD-10 code was present or if a text entry indicated tobacco history, negative if a text entry indicated no tobacco history, and unknown if there was no ICD-code for tobacco history and no indicative text entry value [34]. Overall, we considered a patient to have a positive history of a risk factor only if the clinical variable was documented no later than 2.5 years after the earliest AAA measurement.

### Programming and statistical methods

Data processing was performed in Python 3. The Python wrapper for SQL, SQLAlchemy [35], was used to query electronic medical record data stored in the OMOP Common Data Model. We also used the Python libraries pandas [36] and NumPy [37] for data processing, as well as matplotlib [38] for visualization. For the first-last method, we calculated each patient's total

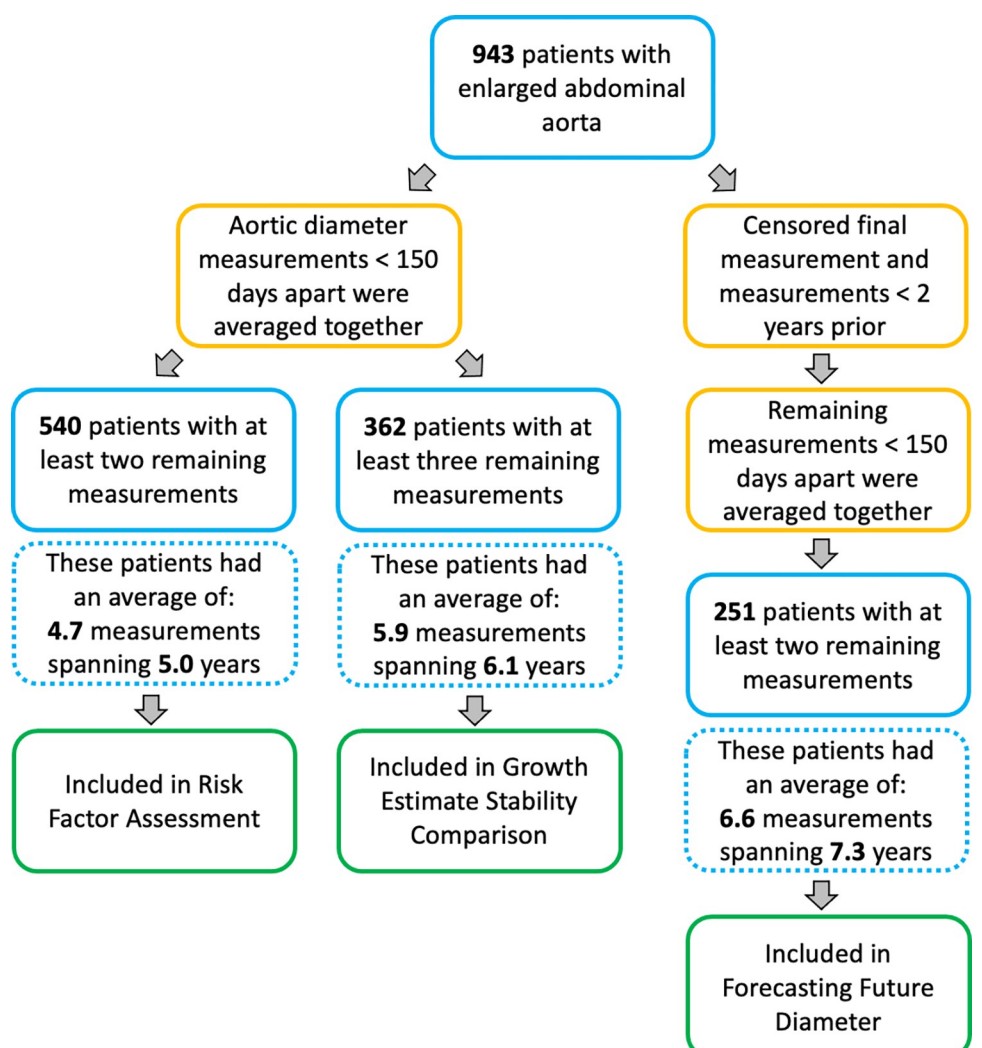

**Fig 1. Study design to assess the stability, predictive accuracy, and statistical power of computational approaches for modeling abdominal aortic aneurysms.** The original dataset was preprocessed and filtered as indicated. For risk factor assessment, only two measurements were needed for each patient. After combining close-together measurements, 540 patients had at least two measurements. For the growth estimate stability comparison, a minimum of three measurements were needed for each patient. After combining close-together measurements, 362 patients had at least three measurements. Lastly, for forecasting future diameter, each patient's final measurement was used as the "target" datapoint to predict, and therefore censored from the models' training data. To ensure that the forecasting period represented a substantial gap in time, measurements less than two years prior to the target point were also censored. After censoring these datapoints, close-together measurements were merged, resulting in 251 patients with two or more measurements in the training dataset. The dataset summaries in the dashed boxes reflect the quantity of the raw data, i.e. before averaging close-together points and censoring endpoints.

time observed by subtracting their earliest measurement date from their latest measurement date. Their total AAA growth was calculated by subtracting their earliest AAA diameter measurement from their latest AAA diameter measurement. The "first-last" growth rate was then calculated as total growth divided by total time observed. Traditional, unpooled linear regression and unpooled exponential curves were fitted to patients' measurements using the Python library SciPy [39]. For the linear mixed-effects model, we used the statistical platform Stan [40] via its Python interface PyStan [41]. For the exponential mixed-effects model, we used the

statistical platform OpenBUGS [42]. Code for the Stan model and the OpenBUGS model is provided in the S1 Appendix and on Github.

For the exponential models, we developed the formula $y = e^{a(t+b)}+2$, where $y$ is the projected AAA size in centimeters, $t$ is the time in years since the first measurement, and $a$ and $b$ are parameters fit by the model. The addition of two adds the assumption that, at an arbitrarily early time point when $t$ is a large negative number, the estimated aortic diameter should approach the normal size of 2 centimeters. The parameter $a$ controls the shape of the curve, whereas the parameter $b$ allows the curve to be effectively translated left or right, thereby adjusting the curve to fit a wide variety of starting points, since some patients have their aneurysm detected at a smaller diameter and others at a larger diameter. In order to have a single number to compare the exponential growth rates to the linear models, we used the exponential models to estimate the dates at which each patient's AAA would be 4.0 and 4.5 centimeters, and then calculated the linear growth rate between these two points. We selected 4.0 and 4.5 as benchmarks because they lie in the middle of a typical curve. For the downstream analyses, whenever a model returned a negative growth rate, it was replaced with zero.

## Experiment setup

### Methods for growth estimate stability comparison

First, we designed an experiment to test the stability of each model. A reliable, noise-tolerant model should be consistent in its conclusions; adding or removing a datapoint should not drastically change a patient's estimated growth rate. Conversely, a less reliable model might output highly variable growth rates for a single patient, depending on which of the patient's datapoints are input into the model. We were also interested in whether model outputs might show a directional bias, tending to increase or decrease their outputs depending on the time period covered by the input data. This question reflects a real-world problem, since patients' AAAs are not usually observed over their entire natural history. Some patients may have more missing data on the left side, i.e. an AAA that happened to be diagnosed late in its course. Others may have missing data on the right side, i.e. an AAA that has not yet been followed to a large size. To simulate the effect of missing data with the available data, we created two new datasets: one with the earliest datapoint from each patient removed ("left-censored"), and one with the latest datapoint of each patient removed ("right-censored"). In order for each patient to have at least two datapoints in each dataset, we excluded patients with fewer than three datapoints in the original dataset, leaving 362 patients. We then fit each model to all three datasets and compared the estimated growth rates for each patient. In a noise-tolerant model, we expect to obtain similar estimates for a given patient's AAA growth rate across each of the three datasets. If the model is not biased by the direction of censor (removal of the first point versus the last point) then the error should be symmetrical; if it is biased, then we expect to observe reduced growth rate estimates on the right-censored data and inflated growth rate estimates on the left-censored data.

**Methods for forecasting future diameter** Next, we tested whether the models were accurate in predicting a patient's future AAA size, which is an important clinical task. Instead of designing a prospective trial, we simulated "future" data by censoring each patient's last measurement from the model input. Thus, we created a test dataset of each patient's last observed measurement, to be used as the prediction target. We then trained each model on the patients' prior observations, censoring the target point and any measurements less than two years before the target. We excluded patients with fewer than two time points in the training dataset, after merging points < 150 days apart, leaving 251 patients. The average time gap between the last training datapoint and the target point was 3.5 years, and patients had an average of 3.5

observations in the training data. After fitting the models to the training datasets, each model was used to predict each patient's AAA size at the time of the target point. The error was calculated as the predicted size minus the actual target size.

**Methods for risk factor assessment.** Lastly, we wanted to know how the use of different models might influence the assessment of risk factors for AAA growth. To this end, we applied each model to the full dataset of patients with at least two measurements after merging points <150 days apart. We then tested for statistical associations between patients' growth rates and their clinical variables. For categorical variables, we performed Mann-Whitney U tests. For numerical variables, we performed Spearman rank correlations. Both tests were performed with the Python library SciPy. We used a threshold of $p < 0.05$ to suggest whether a researcher would have detected statistical significance if they had relied on a particular model, without adjusting for repeated comparisons.

We hypothesized that some potential risk factors might influence the time of AAA detection or the extent of follow-up, since patients with certain conditions receive more medical attention (affecting the degree of left-censor). Further, older or sicker patients might have a higher surgery risk and thus be followed to a larger size, affecting the degree of right-censor. These factors might lead to bias in the estimate of AAA growth rate, especially in linear models. Thus, to assess the impact of observation period, we also calculated the average AAA size at detection and average age at detection for each categorical variable. For numerical variables, we calculated the strength of association between the clinical variable and the starting size and age at detection.

## Results

### Applying the models

We applied five growth metrics to patients' longitudinal AAA data and evaluated the clinical utility of each model. We successfully applied all five modeling methods to the datasets for each experiment (Fig 1). In the unpooled exponential model, the optimizer occasionally failed to fit parameters for certain patients. In these cases, we substituted the estimate from the linear regression model. This failure typically occurred when the linear regression model assigned a negative growth rate or a growth rate very close to zero, indicating a situation where a curve fit was not feasible. For the hierarchical mixed models, trace plots were constructed to assess convergence between chains. No issues with convergence were noted in the linear hierarchical mixed model. In the exponential hierarchical mixed model, the chains failed to converge for 1–2% of patients, depending on the dataset. Regardless of convergence, parameters were taken from the central tendency of the values across the chains.

### Relationship between estimated growth rate and initial AAA size

As a baseline characterization of each model, we calculated the median of the growth rates from each model. Table 1 shows the median growth rate assigned by each model on the full dataset of 540 patients. The distribution of growth rates varied considerably across models. The unpooled linear model assigned the lowest median growth rate of 1.47 millimeters per year, and the exponential mixed model assigned the highest median growth rate of 1.89 millimeters per year. In addition to having different median growth rates, the models also varied in their relationship to starting size. The growth rates from the linear models were highly correlated with starting size; patients with larger AAA diameters at detection were assigned higher growth rates (Fig 2). In the first-last model and the unpooled linear model, growth rates increased monotonically with starting size groups; the same was true for all but the last size category in the linear mixed model. This relationship was highly significant according to the

**Table 1. Median AAA growth rate according to each model and relationship between growth rate and initial diameter.**

| Model | Median growth rate (mm/year) across 540 patients | Correlation between growth rate and initial diameter (Spearman R) | P-value of correlation (Spearman R) |
|---|---|---|---|
| First-last | 1.57 | 0.404 | p < 0.001 |
| Linear unpooled | 1.47 | 0.403 | p < 0.001 |
| Linear mixed model | 1.60 | 0.456 | p < 0.001 |
| Exponential unpooled | 1.81 | 0.030 | p = 0.48 |
| Exponential mixed model | 1.89 | 0.233 | p < 0.001 |

Spearman rank test (Table 1). In contrast, the exponential mixed model showed a much smaller correlation coefficient between growth rate and starting size (0.23 vs 0.40–0.46 for the linear models), although the relationship was still statistically significant. For the unpooled exponential model, there was no detectable relationship between growth rate and starting size. These results suggest that the exponential models were much less biased by the observation window, which is further explored in the following experiment.

**Growth estimate stability comparison.** Next, we evaluated model stability and directional bias in response to missing data. In theory, a reliable model should produce consistent estimates of growth rate in the same patient, regardless of the time points that happen to be observed for that patient. To simulate missing data, we created a test dataset by removing each

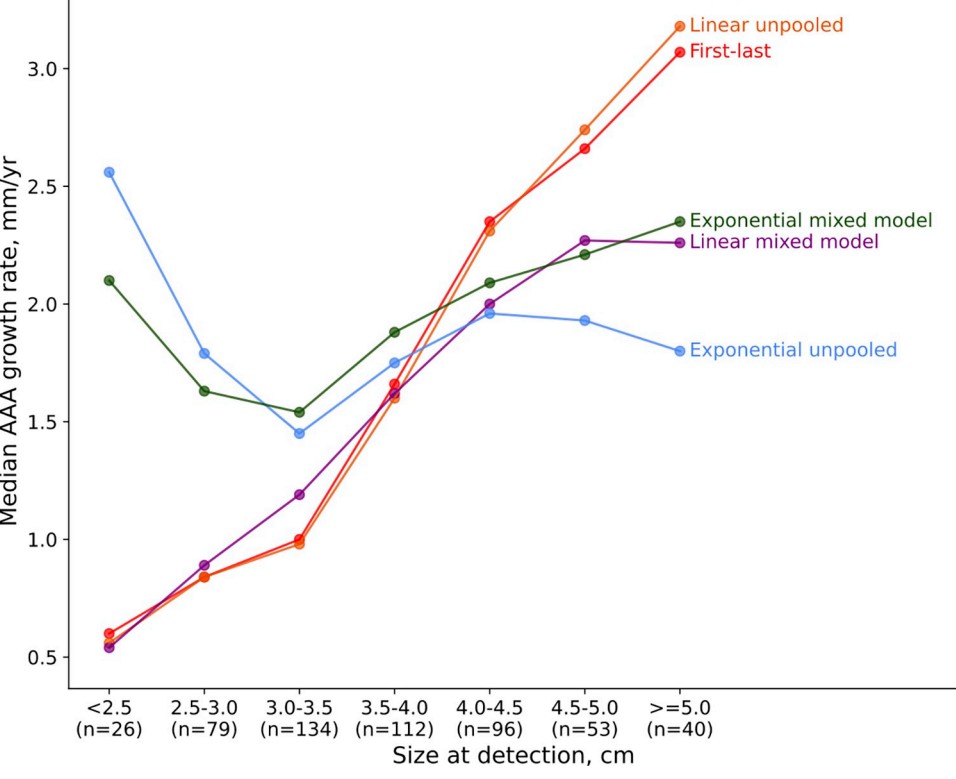

**Fig 2. Impact of AAA diameter at detection on estimated AAA growth rate.** The linear models showed a clear relationship between size at detection and growth rate, in which AAAs discovered at larger sizes were estimated to be faster growing. In the exponential models, however, a larger size at detection did not necessarily imply a higher growth rate. (n = number of patients).

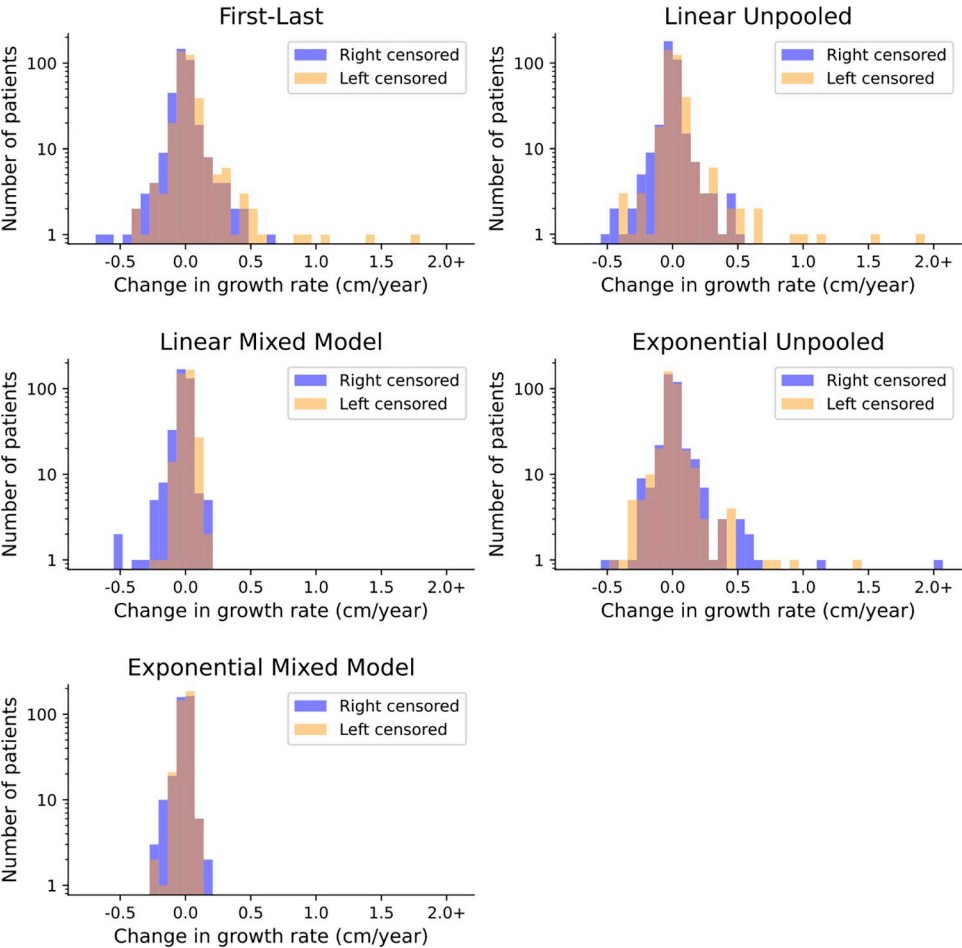

**Fig 3.** Impact of the removal of the earliest (left-censored) or latest (right-censored) aortic diameter measurements on estimation of aneurysm growth rates. We evaluated the magnitude and direction of changes in growth rate estimates from each model. A wider-spread histogram was seen in the unpooled models, meaning more patients had large changes in their growth rate estimate, indicating that the model was unstable and sensitive to noise. The two mixed models showed a narrower distribution, indicating stability and noise tolerance. A similar distribution of changes from left-censoring the data (orange) and right-censoring the data (blue) was seen in the exponential models, indicating that the direction of change was unrelated to the direction of censor. In the linear models, the left-censored distribution was shifted right, meaning the model tended to assign higher growth rates after losing the earliest datapoint, and vice versa. This asymmetry suggested that the linear models were more vulnerable to bias from the observation window, such as assigning higher growth rates to aortic aneurysms detected at large sizes, and assigning lower growth rates to aortic aneurysms not yet followed to a large size.

patient's first AAA measurement ("left-censored") or last measurement ("right-censored.") In certain models, the removal of a single datapoint caused dramatic changes, whereas other models were more stable. Namely, the first-last, unpooled linear regression, and unpooled exponential models showed substantial instability, represented by a broad spread in the distribution of changes (Fig 3). For some patients, growth rate estimates increased by more than 1 centimeter per year after removal of a single datapoint. In contrast, the linear and exponential mixed models were relatively stable, showing few extreme changes in patients' estimates. The distribution of changes in patients' growth rates in these models was more narrow, with most changes near zero. In addition to the magnitude of changes, we were also interested in whether the errors were symmetrical. We expected linear models to increase their estimates when early data was lost, and decrease them when late-stage data was lost. Indeed, all the linear models

**Table 2. Differences between projected and actual AAA diameters.**

| Model | Mean of squared projection errors[a] (+cm$^2$) | Median of absolute* projection errors[a] (+cm) | Mean of absolute* projection errors[a] (+cm) | Mean of raw[+] projection errors[a] (+/-cm) | Mean of mean squared errors to training data[b] (+cm$^2$) |
|---|---|---|---|---|---|
| First-last | 0.758 | **0.290** | 0.474 | -0.057 | 0.01324 |
| Linear unpooled | 0.748 | 0.294 | 0.468 | -0.053 | 0.00814 |
| Linear mixed model | **0.297** | 0.300 | 0.406 | -0.150 | 0.01286 |
| Exponential unpooled | 49274.202 | 0.309 | 15.896 | 15.480 | **0.00782** |
| Exponential mixed model | 0.312 | 0.293 | **0.389** | **0.002** | 0.01735 |

[a]Projection error (all but the final column) is error in predicting the "future" point that was withheld from the models.

[b]Error to training data (shown in the final column) is distance between the models and the training points for each patient.

*Absolute error indicates magnitude of error, regardless of direction, i.e. absolute value of error. A smaller value indicates that the model tends to be close to the actual value, regardless of whether the error is an overestimate or an underestimate.

[+]Raw error indicates directional error, which includes the error's sign (+/-). Overestimates and underestimates cancel out. If the mean raw projection error is negative, the model tends to underestimate future aortic diameter.

The value closest to zero in each column is bolded.

showed some directional bias, meaning they assigned decreased estimates of growth rate on the right-censored dataset and increased estimates on the left-censored dataset. The unpooled exponential model and exponential mixed model, however, showed a relatively symmetrical error. Again, this result suggests that the exponential models suffer less bias from the observation window. Notably, the only model that showed both stability and symmetry was the exponential mixed model.

**Forecasting future diameter.** Prediction of future aortic diameter is an important clinical task, because expectations about a patient's future size are used to determine safe follow-up intervals and to inform decisions about surgical intervention. In our forecasting experiment, we simulated this clinical task by withholding each patient's final measurement from the data used to fit the models. We found substantial variation in the ability of the models to accurately forecast patients' final AAA measurement, as well as their tendencies to overestimate or underestimate (Table 2). A typical measure of error is mean squared error, which inflates the penalty against large errors. Considering the mean of the squared errors between predicted and actual measurement ("projection error") revealed that the linear mixed model had the smallest error, closely followed by the exponential mixed model (about 0.3 cm$^2$). The unpooled linear model and first-last model had substantially higher error (about 0.75 cm$^2$). The error from the unpooled exponential model was orders of magnitude larger, which was due to a small number of extreme values. Overall, in terms of the mean squared error metric, the mixed models showed a clear advantage.

To reduce the penalty to outliers, we also considered the median size of the error, still removing the effect of the error's sign by taking the absolute value instead of the square ("absolute" projection error). This metric was similar across the models (0.29 cm to 0.31 cm), suggesting that the differences in mean squared error were mostly driven by the tails of the error distributions. Clinicians may also want to know the average magnitude of the error in centimeters, i.e. how far, on average, the prediction will be from the actual future size. This metric (the mean of "absolute" projection errors) was lowest in the exponential mixed model at 0.389 centimeters, but was only slightly higher in the linear models. Again, the metric was orders of magnitude higher in the unpooled exponential model, further illustrating that this model produced some extreme outliers.

We were also interested in the models' tendencies to overestimate or underestimate future size, which we measured by averaging the models' prediction errors, while retaining the sign of the error instead of using the absolute value ("raw" projection error). Across all the linear models, the predicted measurement was an average of about one millimeter less than the actual measurement, suggesting a tendency to underestimate future size. In the exponential mixed model, however, the mean of the errors was very close to zero, suggesting that the exponential mixed model achieved the best balance of overestimates and underestimates.

## Model closeness of fit to training data

To examine how models interact with training data, it is important to consider the degree of error between a model and its training data. Minimal error relative to the training data indicates that a model is able to fit the known datapoints very closely, whereas higher training error indicates a less optimized fit. However, there can be a trade-off between training error and testing error, where a model that "over-fits" the training data tends to lose external validity. For our models, we found the lowest training error in the unpooled exponential model, followed by the unpooled linear model. The training error was higher in the first-last method, which makes sense considering that this method does not consider all of the datapoints for patients with more than two datapoints. The training error was also higher in the mixed models, which reflects the hierarchical nature of these models. Namely, they select parameters that strike a balance; they seek parameters that fit the individual patient's datapoints, but also seem probable given the overall distribution of parameters being learned from the cohort as a whole. Therefore, it follows expectation that the models with lower training accuracy had higher accuracy in predicting the censored, final measurement. Ultimately, the exponential mixed model had the largest error relative to the training data, low error in prediction, and the best balance between over-estimating and under-estimating future size.

## Risk factor assessment

Lastly, we examined the association of patient-level risk factors, such as smoking history and diabetes status, with fast AAA growth. Identification of risk factors is important for making predictions at the patient level, as well as determining possible interventions, such as smoking cessation. Therefore, a model with greater statistical power to detect risk factors may be preferred by clinicians and researchers. Thus, we evaluated whether different models might detect different risk factors for AAA growth rate. We excluded the unpooled exponential model from this experiment due to its large error in the forecasting experiment. Across the remaining four models, we found some variation in the number of clinical variables detected as risk factors, as well as the strength of the associations. Among the categorical variables, five clinical factors appeared statistically significant ($p<0.05$) or borderline ($p<0.1$) in all four models: diabetes, hypertension, cerebrovascular disease, coronary artery disease, and chronic kidney disease (Table 3). Diabetes was found to be a strong negative risk factor according to all the models ($p<0.03$ or less). Interestingly, the mixed models produced a more confident result for diabetes ($p<0.001$). However, the exponential mixed model also reported a smaller effect size. According to the linear models, AAAs in patients with diabetes grew about 0.5 mm slower per year than patients without diabetes. In the exponential mixed model, the difference was halved, at just 0.25 mm slower. In all four models, five categorical variables were not associated with AAA growth rate: race, metformin use in patients with diabetes, insulin use in patients with diabetes, statin use, and chronic obstructive pulmonary disease (Table 4). It should be noted that some of these categories suffered from small sample sizes, particularly non-white race, metformin use, and insulin use. Lastly, three variables were reported as significant or

**Table 3. Clinical variables that were significantly associated (p < 0.1) with average growth rate by all four models.** Average growth rate is shown in mm/year for each clinical category.

| Diabetes | Present (n = 121) | Absent (n = 375) | Difference | p value |
|---|---|---|---|---|
| First-last | 1.69 | 2.17 | -0.48 | **0.028** |
| Linear unpooled | 1.65 | 2.15 | -0.5 | **0.02** |
| Linear mixed model | 1.43 | 1.83 | -0.4 | **<0.001** |
| Exponential mixed model | 1.74 | 1.99 | -0.25 | **<0.001** |
| Hypertension | Present (n = 427) | Absent (n = 69) | Difference | p value |
| First-last | 2.16 | 1.38 | 0.78 | **0.005** |
| Linear unpooled | 2.14 | 1.35 | 0.79 | **0.002** |
| Linear mixed model | 1.8 | 1.34 | 0.46 | **0.007** |
| Exponential mixed model | 1.96 | 1.76 | 0.2 | **0.041** |
| Cerebrovascular disease | Present (n = 95) | Absent (n = 401) | Difference | p value |
| First-last | 1.67 | 2.14 | -0.47 | **0.035** |
| Linear unpooled | 1.63 | 2.13 | -0.5 | **0.028** |
| Linear mixed model | 1.5 | 1.79 | -0.29 | 0.084 |
| Exponential mixed model | 1.81 | 1.96 | -0.15 | 0.071 |
| Coronary artery disease | Present (n = 366) | Absent (n = 130) | Difference | p value |
| First-last | 2.19 | 1.66 | 0.53 | **0.033** |
| Linear unpooled | 2.17 | 1.63 | 0.54 | **0.026** |
| Linear mixed model | 1.8 | 1.55 | 0.25 | 0.058 |
| Exponential mixed model | 1.97 | 1.83 | 0.14 | 0.06 |
| Chronic kidney disease | Present (n = 164) | Absent (n = 332) | Difference | p value |
| First-last | 2.25 | 1.95 | 0.3 | **0.035** |
| Linear unpooled | 2.24 | 1.93 | 0.31 | **0.031** |
| Linear mixed model | 1.95 | 1.63 | 0.32 | **0.028** |
| Exponential mixed model | 2.07 | 1.86 | 0.21 | **0.016** |

Average growth rate (mm/year) is shown for each clinical category. The difference was calculated as the second group's average growth rate minus the first group's average growth rate. The p value corresponding to the difference between groups was calculated from the Mann-Whitney U test. p values less than 0.05 are bolded.

borderline according to the exponential mixed model, but insignificant in the other models: female gender, tobacco history, and history of coronary artery bypass graft surgery (Table 5). According to the exponential mixed model, female gender and tobacco history were both associated with faster growth, whereas coronary artery bypass graft surgery was associated with slower growth. Overall, the exponential mixed model detected the greatest number of risk factors.

In addition to statistical sensitivity, we were interested in the potential impact of the observation window on the associations with potential risk factors. Previous experiments showed that linear models are heavily influenced by starting size. Many clinical risk factors may influence the likelihood of screening tests, potentially causing earlier or later detection and therefore different starting sizes. To examine potential effects, we compared patient starting diameter and age at detection by calculating the average values in each clinical category (S1 Table). Patients with diabetes had a significantly smaller size at detection than patients without diabetes. Patients with diabetes were also an average of 1.5 years younger at detection than patients without diabetes, although this difference only bordered on statistical significance. Also bordering on significance was a smaller detection size in female patients and in patients with cerebrovascular disease. In addition to the categorical variables, we also tested associations with several numerical variables, such as hemoglobin A1C, blood lipid levels, and

**Table 4. Clinical variables that were not significantly associated (p < 0.1) with average growth rate by all four models.** Average growth rate in mm/year is shown for each clinical category.

| Race | White (n = 446) | Other (n = 78) | Difference | p value |
|---|---|---|---|---|
| First-last | 2.05 | 2.09 | -0.04 | 0.489 |
| Linear unpooled | 2.03 | 2.08 | -0.05 | 0.476 |
| Linear mixed model | 1.75 | 1.71 | 0.04 | 0.432 |
| Exponential mixed model | 1.94 | 1.87 | 0.07 | 0.22 |
| **Metformin (diabetics only)** | **Present (n = 14)** | **Absent (n = 107)** | **Difference** | **p value** |
| First-last | 2.06 | 1.64 | 0.42 | 0.274 |
| Linear unpooled | 1.95 | 1.61 | 0.34 | 0.378 |
| Linear mixed model | 1.67 | 1.39 | 0.28 | 0.234 |
| Exponential mixed model | 2.01 | 1.71 | 0.3 | 0.126 |
| **Insulin (diabetics only)** | **Present (n = 9)** | **Absent (n = 112)** | **Difference** | **p value** |
| First-last | 1.15 | 1.73 | -0.58 | 0.101 |
| Linear unpooled | 1.19 | 1.68 | -0.49 | 0.138 |
| Linear mixed model | 1.18 | 1.45 | -0.27 | 0.305 |
| Exponential mixed model | 1.7 | 1.75 | -0.05 | 0.486 |
| **Statins** | **Present (n = 128)** | **Absent (n = 368)** | **Difference** | **p value** |
| First-last | 2.29 | 1.97 | 0.32 | 0.45 |
| Linear unpooled | 2.3 | 1.94 | 0.36 | 0.386 |
| Linear mixed model | 1.83 | 1.7 | 0.13 | 0.343 |
| Exponential mixed model | 1.99 | 1.91 | 0.08 | 0.357 |
| **Chronic obstructive pulmonary disease** | **Present (n = 210)** | **Absent (n = 286)** | **Difference** | **p value** |
| First-last | 2.15 | 1.98 | 0.17 | 0.154 |
| Linear unpooled | 2.15 | 1.95 | 0.2 | 0.138 |
| Linear mixed model | 1.79 | 1.69 | 0.1 | 0.424 |
| Exponential mixed model | 1.96 | 1.91 | 0.05 | 0.282 |

Average growth rate (mm/year) is shown for each clinical category. The difference was calculated as the second group's average growth rate minus the first group's average growth rate. The p value corresponding to the difference between groups was calculated from the Mann-Whitney U test.

blood pressure (S1 Table). These results were mostly non-significant across the models, with the exception of hemoglobin A1C among diabetic patients. Higher A1C was associated with slower AAA growth rate; this effect was significant ($p<0.05$) in the unpooled linear model, but borderline to non-significant in the other models. Overall, the hierarchical mixed models reported more statistically significant effects, and the exponential mixed model reported the greatest number of significantly associated variables.

## Discussion

A wide variety of statistical models have been applied in the literature to characterize AAA growth rates. The estimated growth rates from these models have been used to detect risk factors, test interventions, determine optimal surveillance windows, and even aid in decisions about surgical repair. Our study examined the suitability of five different methods. We compared their stability, temporal bias, predictive accuracy, and statistical sensitivity. We found that mixed models offer excellent stability and predictive accuracy. Exponential models reduce temporal bias by effectively modeling curves. Exponential mixed models combine all of these benefits, which may explain why our exponential mixed model detected the greatest number of risk factors.

**Table 5. Clinical variables that were significantly associated (p < 0.1) with average growth rate by the exponential mixed model only.** Average growth rate in mm/year is shown for each clinical category.

| Gender | Male (n = 381) | Female (n = 142) | Difference | p value |
|---|---|---|---|---|
| First-last | 2.0 | 2.17 | -0.17 | 0.137 |
| Linear unpooled | 1.98 | 2.16 | -0.18 | 0.137 |
| Linear mixed model | 1.73 | 1.8 | -0.07 | 0.185 |
| Exponential mixed model | 1.89 | 2.03 | -0.14 | **0.02** |
| **Tobacco history** | **Present (n = 293)** | **Absent (n = 29)** | **Difference** | **p value** |
| First-last | 2.02 | 2.57 | -0.55 | 0.121 |
| Linear unpooled | 2.02 | 2.57 | -0.55 | 0.122 |
| Linear mixed model | 1.75 | 1.68 | 0.07 | 0.181 |
| Exponential mixed model | 1.99 | 1.79 | 0.2 | 0.064 |
| **Coronary artery bypass graft surgery** | **Present (n = 130)** | **Absent (n = 366)** | **Difference** | **p value** |
| First-last | 1.89 | 2.11 | -0.22 | 0.151 |
| Linear unpooled | 1.87 | 2.09 | -0.22 | 0.162 |
| Linear mixed model | 1.65 | 1.76 | -0.11 | 0.122 |
| Exponential mixed model | 1.81 | 1.97 | -0.16 | **0.02** |

Average growth rate (mm/year) is shown for each clinical category. The difference was calculated as the second group's average growth rate minus the first group's average growth rate. The p value corresponding to the difference between groups was calculated from the Mann-Whitney U test. p values less than 0.05 are bolded.

Until now, rarely has there been a direct comparison of the different potential methods using the same dataset, especially the potential impact on risk factors. One study attempted a linear, quadratic, and exponential mixed model, but presented risk factors against linear growth rates only [7]. Since AAAs are well-known to grow faster as they grow larger, we suspected that linear descriptions might fail to characterize them fully, and might fall victim to bias, especially when potential risk factors are related to the time of AAA detection. For example, studies have shown that diabetic patients are more likely than non-diabetic patients to receive a variety of diagnostic tests, including chest CTs [43–46]. Correspondingly, we found that diabetic patients had their AAAs detected at a significantly smaller average size than non-diabetic AAA patients. Such a detection bias could effectively cause non-diabetic patients' AAA data to be more left-censored than that of diabetic patients. Our experiments suggest that left-censoring can increase the estimated growth rate of an AAA, which could potentially inflate the AAA growth rates of non-diabetic patients, and thus cause the protective effect of diabetes to be overstated. Indeed, our linear models suggested that diabetic patients' AAAs grow half a millimeter slower per year on average, whereas the exponential mixed model suggested only a quarter of a millimeter difference. However, the mixed models in general tended to show smaller effect sizes, which might be due to the suppression of extreme values that occurs during partial pooling.

While in the case of diabetes, linear bias might lead to overestimating the effect size, other situations could cause a washout effect. For example, female AAA patients tend to be detected at a smaller AAA diameter, but have faster AAA growth rates [19]. If the impact of starting size is uncontrolled, it could lead to underestimating the effect of gender on AAA growth. Such an effect could potentially explain why our exponential mixed model detected female gender as a risk factor for faster AAA growth, when our linear models did not.

Some studies attempt to control the effect of initial size on growth rate by separating the cohort into bands, where each band encompasses a limited range of starting sizes. A limitation of this approach is the reduction in sample size. For example, Solberg et al. divide their cohort of 234 patients into 7 bands, with some bands having fewer than ten patients, and the largest

band having only 87 patients [19]. Also, since there is still variation within each band, unless the band is infinitely narrow, the effect of starting size would in theory be reduced rather than eliminated. Other studies attempt to control the effect of starting size by including it as a variable in the regression formula. A limitation of both this approach and the banding approach is that they are unable to control for any effects from right-censoring, which could occur if patients with certain risk factors were more likely to be indicated for surgery or more likely to be followed for a longer period of time. Our experiments suggest that our exponential model may reduce such effects, but we have not proven that it eliminates them altogether.

Overall, our results suggest that exponential mixed models are feasible and can successfully model curves without sacrificing predictive accuracy or stability. Further, they may wield more statistical power for detecting risk factors and suffer less bias from varying observation windows. However, their added complexity may not be justified in all applications. In particular, small study cohorts with fewer patients may see less benefit from a using a hierarchical model.

At the other extreme is the simplest method, the first-last metric. Our expectations for this metric were poor, given that it ignores data between each patient's first and last measurements. Yet, we found that the forecasting accuracy from this approach was similar to that of unpooled linear regression, and not far behind the other models. The average magnitude of the prediction error was only about a tenth of a millimeter greater in the first-last method than the exponential mixed model, although the mean squared error was substantially higher. In some clinical applications, the simplicity of first-last or unpooled linear regression approaches may be prized over other factors. These methods can be applied to an individual patient in isolation, and do not require advanced statistical software or programming. Notably, the first-last method is extremely accessible; it can even be computed by hand. Our results suggest that although these methods are less accurate, the difference is modest. Still, according to our results, clinicians should be aware that every linear model tended to underestimate future diameter, which errs on the side of inattentive monitoring and thus increased risk for rupture and death.

Further, we note that some of the differences between models that we observed may be dataset-dependent. For instance, many patients in the forecasting experiment had only two datapoints to use for modeling, making first-last and unpooled linear regression equivalent for these patients. Linear regression might outperform the first-last method more noticeably in datasets with many observations per patient. Likewise, the benefit from exponential models may be greater when the observation time is longer, making curves more pronounced.

This study has addressed five mathematical options for modeling changes in maximal aortic diameter, which is the standard metric reported in clinical radiology documents. However, it is important to note that other geometric properties may be used instead of or in addition to aortic diameter. Existing studies have examined the predictive utility of several other aortic properties including eccentricity, tortuosity, undulation, radius of curvature, and spherical diameter [9, 47, 48]. For study teams that have the necessary resources to compute them, these additional metrics may be helpful in increasing the accuracy of AAA growth rate descriptions and predictions.

Overall, we suggest that future studies attempt to validate the methods they use, perhaps by comparing different methods, or assessing accuracy in forecast or goodness of fit. At minimum, studies should state the method they used, as it may be a pivotal detail that could explain puzzling results. For example, a randomized clinical trial of propranolol for AAA treatment found no statistical difference in the primary outcome, AAA growth rate. Yet, participants who received propranolol had lower risk of surgical AAA repair [25]. Given that the decision for surgery is mostly dependent on AAA diameter, it is puzzling that this difference could

occur without a difference in growth rate. Perhaps the method used to assess growth rate might not have characterized differences sufficiently. Such possibilities are difficult to assess, however, when the exact method of growth rate modeling is not stated. Our results indicate that AAA growth rate metric is an important methodological choice, which can affect a study's outcomes. We recommend that researchers make this choice with care.

## Conclusion

We found that linear models of AAA growth rate were subject to directional bias and tended to underestimate future AAA diameters. We also found that unpooled models were sensitive to noise, whereas mixed models were more reliable. The choice of model also affected which risk factors were associated with growth rate; our exponential model detected the greatest number of risk factors. The exponential mixed model also offered excellent pruning of noise while successfully modeling curves. Therefore, exponential mixed models may be an optimal choice for sufficiently large studies. Regardless of the application or method chosen, we recommend that future studies of AAA growth rate should state the method used and provide validation when possible.

## Supporting information

**S1 Appendix.**
(DOCX)

**S1 Table. Association of diameter at AAA detection and age at AAA detection with categorical risk factors.** The average size at detection (in centimeters) and age at detection (in years) is shown for each clinical category. The magnitude of the difference between the groups is also shown. The associated p-value was calculated from the Mann-Whitney U test.
(XLSX)

**S2 Table. Association of aortic aneurysm growth rate with numerical risk factors.** For each clinical risk factor, we calculated the Spearman R correlation coefficient between the clinical factor and the growth rate of each model. We also report the association between the clinical risk factor and initial AAA diameter, as well as the association between the clinical risk factor and age at AAA detection. p-values were calculated to assess whether the relationship is statistically significant.
(XLSX)

## Acknowledgments

The authors thank Kimberly Bacchia for her contributions in developing the code for querying clinical variables, Siao Sun for his influence in the design of the work, and Dr. Amy N. Abell for her helpful critiques of the work's presentation.

## Author Contributions

**Conceptualization:** Kayley Abell-Hart, Janos Hajagos, Victor Garcia, Mary Saltz, Joel Saltz, Apostolos Tassiopoulos.

**Data curation:** Kayley Abell-Hart, Janos Hajagos, Victor Garcia, James Kaan.

**Formal analysis:** Kayley Abell-Hart, Janos Hajagos, Wei Zhu.

**Funding acquisition:** Kayley Abell-Hart.

**Investigation:** Kayley Abell-Hart.

**Methodology:** Kayley Abell-Hart, Wei Zhu.

**Project administration:** Mary Saltz, Joel Saltz, Apostolos Tassiopoulos.

**Software:** Kayley Abell-Hart, Victor Garcia.

**Supervision:** Janos Hajagos, Wei Zhu, Mary Saltz, Joel Saltz, Apostolos Tassiopoulos.

**Validation:** Kayley Abell-Hart.

**Visualization:** Kayley Abell-Hart.

**Writing – original draft:** Kayley Abell-Hart.

**Writing – review & editing:** Kayley Abell-Hart, Janos Hajagos, Victor Garcia, James Kaan, Wei Zhu, Mary Saltz, Joel Saltz, Apostolos Tassiopoulos.

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
