## [Decision Letter · Decision Letter 0]

12 May 2023

PONE-D-22-34426Investigation of commonly used aortic aneurysm growth rate metrics: comparing their suitability for clinical and research applicationsPLOS ONE

Dear Dr. Saltz,

Thank you for submitting your manuscript to PLOS ONE. After careful consideration, we feel that it has merit but does not fully meet PLOS ONE’s publication criteria as it currently stands. Therefore, we invite you to submit a revised version of the manuscript that addresses the points raised during the review process.

We look forward to receiving your revised manuscript.

Kind regards,

Emma Rezel-Potts

Academic Editor

PLOS ONE

a) Did participants provide their written or verbal informed consent to participate in this study?

3. "Your ethics statement should only appear in the Methods section of your manuscript. If your ethics statement is written in any section besides the Methods, please delete it from any other section.

Reviewers' comments:

Reviewer's Responses to Questions

**Comments to the Author**

1. Is the manuscript technically sound, and do the data support the conclusions?

Reviewer #1: Yes

Reviewer #2: Yes

2. Has the statistical analysis been performed appropriately and rigorously? 

Reviewer #1: I Don't Know

Reviewer #2: Yes

3. Have the authors made all data underlying the findings in their manuscript fully available?

Reviewer #1: Yes

Reviewer #2: Yes

4. Is the manuscript presented in an intelligible fashion and written in standard English?

Reviewer #1: Yes

Reviewer #2: Yes

5. Review Comments to the Author

Reviewer #1: Thank you for giving me the chance to revise this very interesting and useful paper.

Introduction:

General comment: The introduction is quite long. Despite that you are analytic and provide many information, some of them should be better placed into the discussion section.

Line 60: think of replacing “between” with “among”

Lines 68-69: This sentence is confusing. Please consider rephrasing.

Lines 100-102: You should better eliminate these lines. They seem rather like a conclusion. The aim of the study should be described at the end of the introduction.

Methods

Lines 117-118: Please explain your abbreviations

Results

Very good and analytic presentation of findings, despite that your analysis is quite complex.

Discussion

Really a great work, especially with the presentations of examples. It becomes quite clear even for someone with less knowledge on statistical models to understand your findings.

Reviewer #2: This study examined various growth models for stability, temporal bias, prediction accuracy, and inferred risk. According to the findings of this study, exponential mixture models have less bias, stability, and risk prediction than other linear models, which has a significant impact on AAA clinical practices. Overall, the approach is solid and the results are well organized and explained.

Major comments

1. The morphological features of abdominal aortic aneurysm shapes are becoming increasingly important for aneurysm growth and rupture due to the increased use of AI and semi-automatic operation (Baek and Arzani, 2022). However, this study only looked at maximum diameters, and other morphological features were not considered for prediction capabilities. Similarly, a new measurement of maximally inscribed sphere diameters was introduced and expanded its use for quantifying aneurysm growth rate and demonstrating growth prediction performance using semi-segmentation. Furthermore, Akkoyun et al. (2021) demonstrated that spherical diameter growth rates provided less bias than transverse maximum diameter measurements in terms of prediction capabilities. It would be interesting to compare it to Akkoyun et al's study in terms of bias, stability, and prediction capability, or at the very least, to discuss other 3D-based morphological features for prediction of growth and rupture risk in terms of limitations and future improvements.

S. Baek and A. Arzani, Current state-of-the-art and utilities of machine learning for detection, monitoring, growth prediction, rupture risk assessment, and post-surgical management of abdominal aortic aneurysms, Applications in Engineering Science, vol. 10, article 100097, 2022

E. Akkoyun, H, Gharahi, S.T. Kwon, B.A. Zambrano, A. Rao, A.C. Acar, W. Lee, S. Baek, Defining a master curve of abdominal aortic aneurysm growth and its potential utility of clinical management, Computer Methods and Programs in Biomedicine, vol. 208, 106256, 2021

Minor comments

2. AAAs have several different maximum diameter measurements. Please specify the maximum diameter used in this study.

3. More information about the final selected data set should be provided, such as imaging modality, average durations, average number of scan imaging per patient, and so on.

6. PLOS authors have the option to publish the peer review history of their article (what does this mean?). If published, this will include your full peer review and any attached files.

Reviewer #1: No

Reviewer #2: **Yes: **Seungik Baek

---

## [Author Response · Author response to Decision Letter 0]

25 Jun 2023

In response to the editors question about our ethics statement: the study was approved by the Stony Brook Internal Review Board. Because study activities were limited to records review, the review board waived consent requirements for this study. This statement now only appears in the Methods section of the manuscript. 

Response to reviewers: 

Reviewer #1: Thank you for giving me the chance to revise this very interesting and useful paper.

Introduction:

General comment: The introduction is quite long. Despite that you are analytic and provide many information, some of them should be better placed into the discussion section.

Thank you for this feedback. We have made the introduction slightly shorter by removing lines 100-102 as suggested below, but we have kept the remaining content in the introduction because the paper’s target audience is diverse, and we want to ensure that we provide all the information necessary to understand the study and its purpose.

Line 60: think of replacing “between” with “among”

Thank you for this suggestion. We have changed “between” to “among.”

Lines 68-69: This sentence is confusing. Please consider rephrasing.

Thank you for this feedback. We have rephrased this sentence.

Lines 100-102: You should better eliminate these lines. They seem rather like a conclusion. The aim of the study should be described at the end of the introduction.

We have deleted these lines.

Methods

Lines 117-118: Please explain your abbreviations

Thank you for pointing this out. We have added the expanded forms for CCS, LOINC, SNOMED, ATC, and ICD-10-PCS. Please note that the abbreviation “OMOP” also appears in these lines but is already defined above. 

Results

Very good and analytic presentation of findings, despite that your analysis is quite complex.

Discussion

Really a great work, especially with the presentations of examples. It becomes quite clear even for someone with less knowledge on statistical models to understand your findings.

Reviewer #2: This study examined various growth models for stability, temporal bias, prediction accuracy, and inferred risk. According to the findings of this study, exponential mixture models have less bias, stability, and risk prediction than other linear models, which has a significant impact on AAA clinical practices. Overall, the approach is solid and the results are well organized and explained.

Major comments

1. The morphological features of abdominal aortic aneurysm shapes are becoming increasingly important for aneurysm growth and rupture due to the increased use of AI and semi-automatic operation (Baek and Arzani, 2022). However, this study only looked at maximum diameters, and other morphological features were not considered for prediction capabilities. Similarly, a new measurement of maximally inscribed sphere diameters was introduced and expanded its use for quantifying aneurysm growth rate and demonstrating growth prediction performance using semi-segmentation. Furthermore, Akkoyun et al. (2021) demonstrated that spherical diameter growth rates provided less bias than transverse maximum diameter measurements in terms of prediction capabilities. It would be interesting to compare it to Akkoyun et al's study in terms of bias, stability, and prediction capability, or at the very least, to discuss other 3D-based morphological features for prediction of growth and rupture risk in terms of limitations and future improvements.

S. Baek and A. Arzani, Current state-of-the-art and utilities of machine learning for detection, monitoring, growth prediction, rupture risk assessment, and post-surgical management of abdominal aortic aneurysms, Applications in Engineering Science, vol. 10, article 100097, 2022

E. Akkoyun, H, Gharahi, S.T. Kwon, B.A. Zambrano, A. Rao, A.C. Acar, W. Lee, S. Baek, Defining a master curve of abdominal aortic aneurysm growth and its potential utility of clinical management, Computer Methods and Programs in Biomedicine, vol. 208, 106256, 2021

Thank you for mentioning this important topic in AAA research. We have added a section discussing the utility of other geometric properties including spherical diameter, with references to these papers.

Minor comments

2. AAAs have several different maximum diameter measurements. Please specify the maximum diameter used in this study.

Thank you for raising this point. We have added a sentence in the methods section providing more detail on how the aortic diameter was measured. 

3. More information about the final selected data set should be provided, such as imaging modality, average durations, average number of scan imaging per patient, and so on.

Thank you for this suggestion. In the methods section, we have added a summary of the types of imaging modalities used in this study (primarily CT and ultrasound). In figure 1, we have added summary information describing the average number of scans and average duration of follow-up for patients in each experiment.

---

## [Editor Report · Decision Letter 1]

11 Jul 2023

Investigation of commonly used aortic aneurysm growth rate metrics: comparing their suitability for clinical and research applications

PONE-D-22-34426R1

Dear Dr. Saltz,

We’re pleased to inform you that your manuscript has been judged scientifically suitable for publication and will be formally accepted for publication once it meets all outstanding technical requirements.

Kind regards,

Emma Rezel-Potts

Academic Editor

PLOS ONE
---

## [Editor Report · Acceptance letter]

4 Aug 2023

PONE-D-22-34426R1 

Investigation of commonly used aortic aneurysm growth rate metrics: comparing their suitability for clinical and research applications 

Dear Dr. Saltz:

I'm pleased to inform you that your manuscript has been deemed suitable for publication in PLOS ONE. Congratulations! Your manuscript is now with our production department. 

Kind regards, 

on behalf of

Dr. Emma Rezel-Potts 

Academic Editor

PLOS ONE